# Evidence for Cognitive Spatial Models from Ancient Roman Land-Measurement

**DOI:** 10.3390/brainsci15040376

**Published:** 2025-04-04

**Authors:** Andrew M. Riggsby

**Affiliations:** Classics, University of Texas at Austin, Austin, TX 78712, USA; ariggsby@mail.utexas.edu

**Keywords:** wayfinding, allocentric, egocentric, route, survey, cognitive maps, cultural evolution

## Abstract

Influential studies in the history of cartography have argued that map-like representations of space were (virtually) unknown in the Classical Mediterranean world and that the cause of this was an absence of underlying cognitive maps. That is, persons in that time/place purportedly had only route/egocentric-type mental representations, not survey/allocentric ones. The present study challenges that cognitive claim by examining the verbal descriptions of plots of land produced by ancient Roman land-measurers. Despite the prescription of a route-based form, actual representations persistently show a variety of features which suggest the existence of underlying survey-type mental models and the integration of those with the route-type ones. This fits better with current views on interaction between types of spatial representation and of cultural difference in this area. The evidence also suggests a linkage between the two kinds of representations.

## 1. Introduction

It has been influentially argued that the Classical Mediterranean world did not make or use maps and that this supposed technological lack was caused by a culture-wide cognitive lack [1,2,3]. These studies observe both the surviving remains of supposed maps and verbal descriptions of spaces. Though the studies in question use different, discipline-specific terminology—“hodological” (or “one-dimensional”) vs. “cartographic” or (“two-dimensional”) representations—the details of their argument make it clear that what they are talking about is essentially the distinction cognitive psychologists often draw between route and survey representations, respectively. That is, they contend, Classical spatial representation is aways route-type (characteristically egocentric-reference, serial, locomotor-derived) rather than survey-type (characteristically non-egocentric-reference, simultaneous, visually-derived) [1,2,3]. I use “non-egocentric,” to include both the narrowly “allocentric”—that is, reference based on a specific point other than the self—and “bird’s-eye” or “coordinate” or “geocentric” systems—that is, ones that operate with respect to an “absolute” framework. With respect to the principal distinction here between route and survey descriptions, the two non-egocentric systems are equivalent. They go on to argue that the supposed facts about representational style directly reflect the kinds of cognitive models available to individuals within the culture. This set of positions will be labeled the Hodological Hypothesis (HH) in what follows.

Some (including myself) have contested the historical part of this hypothesis (α) on the basis of surviving map-like objects [4,5]. However, doubts about the evidentiary base mean that these arguments remain controversial (see [4] (pp. 154–202) for the following examples). For instance, the clearest example of a map is actually from a millennium later and not provably a copy of an ancient artifact; the most common type of objects (survey “formae”) are arguably tabular diagrams rather than spatial maps; many supposed examples are based on ambiguous verbal descriptions rather than actual artifacts. Because of such concerns, the HH view of maps continues to be cited as an important or even a state-of-the-art view in many publications, for instance [6,7,8,9,10,11]. Moreover, while HH proponents continue to accept the cognitive claim (β), opponents have generally not discussed it since they find it moot. As a result, the present paper has two aims. First, it establishes an argument against the idea of a deficiency in ancient spatial cognition (β) on a basis independent of the contested map evidence. Second, beyond the mere co-existence of route and survey cognitive representations in antiquity, it suggests a possible specific interaction between them.

The present paper does this in two ways. First, it exploits a different body of evidence, in which the definitional issues can be avoided: records of land parcels recorded by Roman *mensores* (“land-surveyors” or, more literally, “measurers”) of roughly the first three centuries CE. Second, it addresses itself much more directly to the cognitive aspects of the issue.

## 2. Materials and Methods

It is obviously impossible to conduct experiments with long-dead subjects. However, there have been recent advances in formulating best practices for research areas, such as the history/evolution of cognition, which are characterized by “sparse” and/or “indirect” evidence [12]. Accordingly, the present research incorporates the principles of “making alternative theories explicit” (Section 1), “external consistency with related theories” (Section 4), and “triangulating across forms and sources of evidence” (Section 3 and Section 4). More specifically, it leverages evidence from a fortuitous context which offers a natural quasi-experiment; it compares multiple ancient texts produced in response to the same well-understood task/prompt. We can observe in ancient texts the same kinds of linguistic features examined in modern studies where responses are elicited for scientific purposes, particularly in wayfinding contexts: deictic and directional terms, spatial prepositions, pronouns, motion terminology [13] (pp. 119–160) [14,15,16,17].

The corpus to be examined for these features consists of inscriptions that record legal descriptions of plots of land. Roman *mensores* characterized irregular parcels of land by recording a path defined by a series of landmarks, which could be natural, artificial, or a combination of both; the technical term both for the process and for the resulting text is determinatio. A number of determinationes survive at least in part, typically inscribed on stone or bronze as part of the official record of a legal dispute about the property in question. The specific dataset relied on in this paper was derived as follows. The first stage was a direct examination of all the texts in a modern collection [18] of the known inscriptions that record boundary disputes regulated by the Roman state (that is, the most common context for determinationes) and extraction of the subset of these which include determinationes (complete or partial). Looking at these determinationes showed that they use highly standardized language. The second stage was then to search the standard database of Latin inscriptions [19] for that standard terminology. The database of inscriptions used contains virtually all known inscribed texts. Thus, the “sample” resulting from looking for that conventional language in that large database should, in fact, include all or nearly all determinationes that survive. The total dataset includes 25 instances (see catalog in Table 1; subsequent references of the form “#N” point to item number N in that catalog). This is not enough for formal statistical analysis, but I should point out that it is a fairly large sample by the standards of evidence for the period. Ideally, we might wish to compare these descriptions to the actual paths described, but that is never possible in detail, and in many cases, we have no idea of the actual topographical referents.

## 3. Results

A preserved textbook gives a sample template for how determinationes were to be constructed. “From the hill which is called A, to B river and through that river to creek C or road D and through that road to the base of mountain E, which place is called F, and thence through the ridge of that mountain to the top and through the top of the mountain through the river branch to the place which is called G and thence down to place H, and thence to cross-roads I and thence through the marker of that to the place recording began.” (Hyginus, *De Condicionibus Agrorum* 74C. My translation uses the language of algebraic variables; the Latin in fact literally uses the demonstrative pronoun *ille* “that” in each instance).

### 3.1. Evidence for Route-Type Representations

The actual determinationes in our corpus follow the same general format as the normative pattern.

(a) Sequential language. The basic form is that of a series of landmarks joined by words like *inde* (thence), *dein* (then), *ad* (to), *in* (into), *usque* (on to), *proximus* (next).

(b) An explicit description of the path. They frequently also describe the path taken between landmarks, whether that follows a physical feature (e.g., *per* “through”) or something more notional (*recto rigore*, *rectura*, *recta regione* all = “in a straight line”).

(c) Egocentric deixis. Finally, there are more scattered references which can only be disambiguated if they are taken to presuppose the egocentric reference of someone proceeding along the path described: *dextra* (on the right), *sinistra*, *ad sinistrum* (on the left), *cis* (on this side of), *trans* (on the other side of).

If this were the extent of our evidence on the topic, it would generally support the HH. The terms used refer only to the landmarks (“to X”) or to the pathways between them (“through X”). Nothing hints at the overall shape of the parcel or at the spatial relationships between non-adjacent landmarks. However, this is not all of the evidence.

### 3.2. Evidence for Survey-Type Representations

The texts also include various kinds of information not found in the pattern, from which, I contend, shows quite different cognitive modeling. Though none of the following features are individually standard, they do show the existence of survey-style description in a number of determinationes.

(d) Reference to non-path landmarks. Occasional phrases explicitly and literally locate one of the route landmarks with respect to another site not on the path: *in conspectu* (in sight of, #1), *haud procul* (not far from, #2), *pro* (in front of, #20), *infra* (below, #5), *in flexu* (in the bend, #9–11).

(e) Allocentric deixis. Elsewhere orientation is expressed not from the implicit point of view of a self-traversing the path in order, but explicitly that of another person performing a specified motion (always expressed in the Latin by the dative plural of the present participle, that is “in respect to those who are Xing”): *navigantibus* (sailing, #2), *adscendentibus* (going up, #2), *descendentibus* (going down, #3). Note that this construction converts otherwise egocentric terms like *dexter* (right) and *citra* (this side of) into allocentric expressions.

Though the context is somewhat different than our main sample, I would note in passing a parallel usage elsewhere. Urns containing cremated remains were typically housed in collective tombs where different niches were owned by different people. Texts specifying the location of these frequently refer to the “right” and “left,” but explicitly from the point of view of “persons entering” (*intrantibus*, *introeuntibus*) the complex.

(f) Framework: topography (For languages/cultures in which such systems are the dominant framework across reference contexts, see [20,21]). Perhaps related to the occurrence of participial phrases like “going up” and “going down” are uses elsewhere of adjectives and adverbs meaning “up” and “down.” For instance, #20 makes frequent use of *deorsum* (downward) and *suorsum* (upward). These two terms, as well as the words for “rightward” and “leftward” discussed in (h) below, are spelled in a variety of different ways. Since there is no genuine ambiguity, I have normalized the orthography. Most of these refer to paths along water courses and so mean in effect “downstream” and “upstream.” This simple binary could, in some sense, have been rendered by “right” and “left,” so we should note the one difference. Recall that without further specification of whose perspective is adopted, “left” and “right” assume a self progressing along the specified path in a particular direction. “Up” and “down” are anchored in such a way as to render the point of view irrelevant. There are similar uses of “upper” and “lower” in #4, 8, 23.

(g) Framework: geographic/cosmological (for languages/cultures in which such systems are the dominant framework across reference contexts, see [22]). There is at least one determinatio (#8) which clarifies “left” and “right” by referencing an absolute frame of reference: the cardinal directions. Moreover, #25 refers to cardinal directions, though given the fragmentary state of the text, it is not clear what the function of these references is.

(h) Framework: technical. Finally, if more speculatively, there is one case in which “left” and “right” are likely fixed relative not to absolute reference nor to physical topography, but to a still external framework devised and stipulated by the *mensores* themselves. A text from Spain (#1) uses not only *dextra* (on the right) but *dextroversus*/*sinistroversus* (toward the right/left). These three terms, lumped together, have been taken as evidence of an ego-centric, route-style framework [3], but a competing interpretation calls that into question. There are a couple of key observations here. First, the physical layout of the text is clearly divided into two parts (with a blank line between them and ekthesis of the first line of the second. Second, uses of *sinistroversus* are confined to the first part; uses of *dextroversus* are in the second part. It has been proposed that the right/left references are with respect not to someone following the path but to a (notional) axis dividing the territory in two [23]. Such an axis was a fundamental part of Roman land-surveying practice in other contexts, and it would perhaps not be surprising here. Our only other uses of these terms in this type of text come from a series of very badly damaged texts (#9, 10, 11), which appear to describe a single determinatio in what is now Romania. The usage is compatible with that proposed for the Spanish case, but the texts are too damaged to provide clear evidence.

### 3.3. Summary

In total, 14 of the 25 surviving texts show one or more features characteristic of survey representation. If we take the basic unit of observation as the project rather than the individual inscription, perhaps a better measure, the figures are 12/21. Moreover, five of the texts that do not show any of these features are of a distinctive type. They are inscriptions on individual marker-stones that establish points along the route of a determinatio. They refer only to the location of the immediately following marker in the sequence rather than recording the entire route. Hence, there is much less scope for features which are on any account intermittent.

## 4. Discussion

The HH is certainly intelligible within current understandings of spatial cognition. The route/survey distinction has been well-established in the literature, at least descriptively, since as early as [24], and continues to be a structuring principle for many different kinds of inquiry in the area [25,26]. It is supported by observations that different kinds of representation have distinctive properties in use (e.g., survey representations encode metric information better [27]; switching frame of reference reduces accuracy [28] The inferences from verbal forms to mental representations that are used to support the HH are similar to those offered in experimental studies [13,29,30]. Finally, others have suggested that there are developmental patterns in individuals that have a similar form to what the HH claims for cultural evolution. Some studies have suggested that children develop different forms of representation sequentially [31,32]. By analogy, it is suggested that Classical Antiquity as a whole did not reach the “adult” stage of survey representation [1,2,33].

Nonetheless, the evidence presented here argues against at least two aspects of the HH—the sharp distinction between route and survey modes and the claim that the latter was simply unavailable to persons in Classical Antiquity. In fact, coexistence and interaction of the two modes fits better with the broader picture as currently understood. As for the radical disjunction of route and survey modes, it has long been realized that the two kinds of representation interact with each other. For instance, controlled experiments have shown that people can draw inferences about implied information equally well within or across frameworks [34]. Moreover, individuals appear to acquire route and survey information simultaneously during experimental tasks [35,36] for possible neural mechanisms [37]. As for the purported parallel with developmental patterns, it has been shown that this developmental pathway is not actually universal, and that its presence maps to language families, not to levels of technological development [13] (pp. 129–131). It would, thus, have been a startling result if ancient Greeks and Romans indeed lacked survey representations, but the detailed evidence shows that is simply not the case.

A few words may be in order about the limitations of the specific data examined here. Land-surveying was an established profession with formalized procedures and, thus, the linguistic features described above might in some sense be an artifact of a very restricted community or use-context. Two considerations militate against that concern. First, we should note the diversity of expression among different determinationes. No single pattern formula could account for all of the sample. Yet, any given expression is typically used in more than one text, and the repertoire of kinds of expression is fairly compact, mostly though not entirely restricted to the nine categories recorded here. This kind of rough family resemblance is more typical of Roman information handling than would have been a precise formalization [4], thus, it is not a sign of professionalization. Second, this diversity appears precisely in the actual determinationes rather than in the normative pattern form. That is, it looks like the more complex representation arises from individual application rather than professional formalization. In fact, we know that experimentally elicited non-specialist descriptions often combine route and survey perspectives without explicit coordination of the two [34]. Combined with the cartographic evidence [4] (pp. 180–194), this suggests that Roman spatial cognition had the same internal variety as we see in contemporary societies. Finally, rather than being a limitation, this situation suggests (at least tentatively) something about the interaction between representations. Despite a formal context which explicitly elicits only route information, the majority of texts also offer survey information. This suggests that the two are not merely available simultaneously, but perhaps inherently linked to each other.

## 5. Conclusions

Evidence from the records of ancient Roman surveyors argues for the co-existence of underling route- and survey-type cognitive spatial representations. This is contrary to considerable opinion in the historical literature (which questions the existence of the latter in the classical Mediterranean world) but concords better with the current scientific literature (which would find such a local limitation unlikely). Beyond mere co-existence of the two representations, the specific elicitation context here suggests actual linkage between the two cognitive representations.

## Figures and Tables

**Table 1 brainsci-15-00376-t001:** Catalog and features of surviving determinationes. EDCS# is the entry number within the “Epigraphik-Datenbank Clauss/Slaby” [19]. Columns (a)–(h) summarize the results of Section 3 by identifying which of the named features occur in which of the determinationes indicated by “x.”

Catalog #	EDCS #	a. Sequential Language	b. Described Path	c. Egocentric Deixis	d. Non-Path Landmarks	e. Allocentric Deixis	f. Topographic Framework	g. Absolute Framework (Natural)	h. Absolute Framework (Artificial)
1	03700456		x	x	x				x
2	08200632	x	x		x	x	x		
3	08200631	x		x	x	x			
4	05801856	x	x	x			x		
5	27100345	x	x		x	x	x		
6	51200146	x	x						
7	07400303	x	x	x				x	
8	20600177	x					x	x	
9	09200012	x	x		x				x
10	30100820	x	x		x				x
11	11600210	x	x		x				x
12	33800076		x						
13	28600127		x						
14	62101299		x						
15	62101300		x						
16	17700548	x	x						
17	19500071	x	x						
18	70800108	x	x						
19	11300715	x		x					
20	20000142	x	x		x		x		
21	14803826	x		x					
22	20403626								
23	36900084			x			x		
24	16201630	x	x		x				
25	12500326	x						x	

In three instances (9–11 14–15, 17–18), multiple inscriptions appear to belong to the same project of determinatio. These are noted by placing the rows for those inscriptions together and shading them. For explanations of the categories in the columns, see Section 3. For a fuller explanation of the broader route/survey distinction, see Section 4.

## Data Availability

Data are publicly available from [19]. Reference IDs for the relevant entries in this database are given in Table 1.

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
