# Peer review of "Evidence for Cognitive Spatial Models from Ancient Roman Land-Measurement"

_brainsci, 2025, doi:10.3390/brainsci15040376_

Round 1
Reviewer 1 Report
Comments and Suggestions for Authors
This paper has a lot of virtues: it is short, readable, and on an interesting topic.
Still, I have a few suggestions and concerns, which are listed below.
Depending on the answers to these concerns, the paper could require anything from minor tweakin to major revisions, or even a transfer to a different journal.
-
There's not much brain science in the paper, but I don't think the paper would benefit from expansion in this direction. At the same time, the paper might be viewed as discussing matters from within the broader cognitive science camp. So, the question is, does the paper's topic adequately fit the journal's scope? This is a question that the journal's editor must decide on. If the paper doesn't fit the scope, hopefully the editors can transfer it to another journal.
-
I have some concerns that a "straw man" argument has been set up. The article opens with the claim that "It has been influentially argued that the Classical Mediterranean world did not make or use maps and that this supposed technological lack was caused by a culture-wide cognitive lack." In the same paragraph, the article goes on to assert: "Classical spatial representation is always route-type (characteristically egocentric-reference, serial, locomotor-derived) rather than survey-type (characteristically non-egocentric-reference, simultaneous, visually-derived)."
The first statement has three citations, two of them from the same author. The second statement has no citation, but maybe the reader is meant to assume the views expressed belong to the authors from the first citation. Whatever the case, more time needs to be spent establishing these opening points. More citations should be offered. Moreover, even if one hundred authors are cited, this does not logically mean that there can't be one thousand authors who share the view of the writer of the submitted manuscript, a scenario in which the case in the manuscript loses its central justification.
-
The authors' work is based on ancient written descriptions, but I want to ask: are there really no actual maps from the Roman world? I did not do a deep delve, but a cursory look indicates quite a few maps from the Roman and Greek world, and other cultures besides. Why are actual maps not discussed? Assuming that I am missing something but it would perhaps help if the rationale was very clearly laid out. Lines 41-44 offer something that could be contrued as a justificaion but to me it is more of an excuse (though the mention of question about what definitionally counts as "maps" is interesting in its own right).
-
Some additional time ought to be spent explaining the justification for the two categories on the chart (route-type vs survey-type). And do I read the manuscript right when I interpret the author as later suggesting that the two categories are not entirely distinct? There's a line of thinking from phenomenology that would hold that, at the very least, the second kind of knowledge depends on the first kind. For a more on this, see the last few pages of Crippen's (2015) Pictures, Experiential Learning and Phenomenology, which has an example involving maps.
-
Another issue could be that the data in the charts has less to do with whether people in the Classical Mediterranean had cognitive maps than with matters of cognitive convenience and technical wherewithal. A person can have a survey-style cognitive map but express things in route-like sequences simply because it is easier and more convenient. Indeed, the route-like description could even be based on a survey-style understanding.
Reviewer 2 Report
Comments and Suggestions for Authors
The manuscript “Evidence for Cognitive Spatial Models from Ancient Roman Land-Measurement” challenges – apparently popular - belief that Classical Mediterranean societies lacked survey-type (allocentric) cognitive spatial models, using evidence from Roman land measurements (determinationes) conducted by land-surveyors (mensores). The author cites some sources which put forward this notion that ancient Mediterranean cultures only used route-type (egocentric) representations for spatial cognition (the “Hodological Hypothesis”). For full disclosure, I have to say that I am NOT a cognitive psychologist – which might explain why I have never heard about this claim that the cognitive models of a highly developed and relatively recent human population might have been limited to sequential, egocentric descriptions, and that they were incapable of creating comprehensive survey-type (allocentric) mental maps. The idea certainly makes no sense in light of evolutionary and developmental neurobiology, nor when we look at comparative research and the complex spatial maps of animals with significantly simpler cortices than those possessed by the ancient Romans. For this reason, I would recommend that the author elaborate more thoroughly on the original claims he is opposing and perhaps cite more than just three references that are between 20 and 40 years old.
In the methodological section, could you please describe more precisely why were these two specific sources of documents chosen (i.e. How many potential sources are there? Are these the only available ones, or are they all that exist on this topic? What were the criteria for selecting these particular databases if more than these two were available?).
Is there a possibility that, because all the documents were written by members of a specialized profession (mensores), their formalized procedures might differ from broader, everyday spatial cognition among the general population? Furthermore, would it be feasible to conduct a similar analysis of – for example - architectural plans or descriptions, especially considering that the Romans left behind impressive structures whose three-dimensional complexity required highly sophisticated mental representations. Additionally, comparing such architectural evidence with land-surveying texts could reveal whether different domains (e.g., architecture vs. land measurement) exhibited similar or distinct cognitive approaches (or maybe just terminology).
All that being said, I did enjoy the paper; I found the approach interesting and innovative. Although I was never aware of the rumors about ancient Romans’ spatial deficiencies, I am glad to see a paper challenging such a ridiculous notion.
Reviewer 3 Report
Comments and Suggestions for Authors
In the manuscript, the author reported an interesting study that examined mental representations of spatial information in people of ancient Roman. The method used in the study was novel in behavior science. However, the topic of manuscript does not fit the scope of the journal. It is difficult to evaluate the study using the criteria in biological psychology and neuroscience.
Round 2
Reviewer 1 Report
Comments and Suggestions for Authors
As stated in the last review, I think this compact paper is decent. That assessment stands.
My main concern in the first review oriented around with lack neurobiology.
In the revised version, little of substance has been added regarding the brain science.
But if the editors of Brain Sciences are comfortable publishing a piece that doesn't really deal with Brain Sciences, then who am I to argue?
(The journal might also consider slightly expanding it's stated scope).
Comments on the Quality of English Language
Regarding me checking "the language could be improved" box, I pretty much always check this since language can always improved. So, this should be taken to indicate that the piece is not clearly written.
Indeed, one of its virtues all along has been clarity and brevity.
Reviewer 3 Report
Comments and Suggestions for Authors
The revision has improved. However, unfortunately, the method of the study did not follow the hypothesis testing protocol, and the topic of the paper may not fit the scope of the journal.